

# Manifestations of muscle fatigue in baseball pitchers: a systematic review

Richard Birfer[1], Michael WL Sonne[2] and Michael WR Holmes[1]

[1] Department of Kinesiology, Brock University, St. Catharines, ON, Canada
[2] Baseball Development Group, Toronto, ON, Canada

## ABSTRACT

**Background:** Fatigue in baseball pitchers is a process linked to lowered physical and mental performance, injury, and changes in kinematics. Numerous studies have associated fatigue with overuse, high ball velocities, lack of rest time, poor mechanics, and degree of self-satisfaction. The aim of this study was to systematically review the literature to identify a theoretical framework for the relationship between outcomes and the manifestation of fatigue on baseball pitching. The synthesized data may identify areas requiring further research.

**Methodology:** This protocol was registered with PROSPERO (ID: CRD42018114194). SPORTDiscus, Medline, PubMed, Cochrane Database of Systematic Reviews, and Google Scholar were searched, using keywords such as fatigue in pitchers and changes in kinematics (e.g., pitching mechanics, valgus elbow torque), performance (e.g., pitch count, pitch type), and injury (e.g., pain, elbow, and shoulder soreness). Three reviewers independently screened the articles, selected relevant literature based on abstract eligibility, and assessed the methods described therein for final inclusion.

**Results:** A total of 31,860 articles were screened for eligibility and 25 articles were included for the review. The selected articles included epidemiological, longitudinal, experimental, conference papers, and crossover laboratory studies. Evidence extracted from the 25 studies demonstrates a relationship between fatigue in baseball pitching, and three overarching outcomes: changes in kinematics, a decrease in performance, and an increase in injury risk.

**Conclusions:** Findings show that a co-dependence between changes in kinematics and a decrease in performance, which stems from central and peripheral fatigue, is a contributing factor of injury in baseball pitchers. A large percentage of baseball pitchers exhibit pain or soreness in either their elbow or shoulder, or both at some point in a season. Initially, kinematic changes occur that could maintain performance, but may increase joint and tissue loading. Performance decreased with elevated pitch counts and innings thrown, and pitching further into games or the season. Evidence was found to be consistent across all studies; however, more work is needed in the area of fatigue as an injury mechanism during pitching. With a proof of concept established, the prevention of negative outcomes associated with fatigue must be the focus of future research and performance should not be the only criteria.

Corresponding author
Michael WR Holmes,
mholmes2@brocku.ca

## INTRODUCTION

The sport of baseball is commonly known as America's pastime. With its continued growth in participation since 2011 (*Outdoor Participation Report, 2018*), baseball is now played year-round. Pitchers of all ages often throw a large number of pitches throughout the calendar year (*Fazarale et al., 2012*), which can result in the gradual accumulation of fatigue (broadly defined as a decrease in force generating capacity), if proper rest and recovery is not considered (*Lyman et al., 2001*). Developmental baseball associations have set maximum pitch count recommendations for specific age groups, although an Internet-based survey confirmed that 27% of youth baseball coaches fail to follow these guidelines (*Fazarale et al., 2012*). Numerous studies have further identified overuse (*Makhni et al., 2014*), high velocities (*Freeston et al., 2014*), lack of rest time (*Crotin et al., 2013*), and pitch type (*Lyman et al., 2002*), among others, as predictors/risk factors for fatigue; all of which are seemingly linked to kinematics, performance, tissue stress and injury.

The cumulative loading caused by fatigue can result in microtrauma, which over time, can contribute to the high prevalence of injury. One of the more prevalent injuries in baseball is a sprain to the ulnar collateral ligament (UCL) (*Yang et al., 2016*), with recovery times for complete tears averaging 20.5 ± 9.72 months after UCL reconstruction (*Erickson et al., 2013*). In the 2018 season, there were 86 cases of Tommy John surgeries across professional baseball (*Roegele, 2018*). An epidemiological study showed that 46% of youth pitchers were encouraged to throw through arm pain, while 82% of players reported arm fatigue during games and practices (*Makhni et al., 2014*). With decreased time between pitches, fatigue reduces overall elbow joint stiffness, which can theoretically lead to increased stress on the UCL (*Sonne & Keir, 2016*). Studies have demonstrated that maximal elbow valgus torque is produced during the arm cocked phase of pitching, when maximum shoulder external rotation is reached (*Yang et al., 2016*). At this point, a group of muscles, the flexor-pronator mass, is a major contributor to providing the elbow with the stability necessary to reduce stress on the UCL (*Sonne & Keir, 2016*).

Muscle fatigue is a process that occurs due to central and/or peripheral mechanisms, which can emerge due to numerous decrements from motor centers to the muscle fiber (*Davis, 1995*) and typically manifests as a decline in maximal force production (*Enoka & Duchateau, 2008*). Changes in kinematics stemming from the onset of fatigue have been well documented. A repeated-measures design was conducted on 16 healthy collegiate-level pitchers in which a fatigue protocol was introduced (*Tripp, Yochem & Uhl, 2007*). Following the fatigue protocol, it was observed that sensorimotor system deficits recovered within 4 min for the elbow joint and 7 min for the scapulothoracic joint (*Tripp, Yochem & Uhl, 2007*). The study also showed that reproducibility of the glenohumeral segment during the arm-cocked phase of a throw failed to recover within a 10-min period. This emphasizes the importance of sensorimotor acuity, proprioception and endurance in abduction and external rotation (*Tripp, Yochem & Uhl, 2007*). Two studies investigating college pitchers noted a decrease in elbow flexion and an increase in hip flexion with greater season pitch count, contributing to an increased joint load at the shoulder and the elbow (*Grantham et al., 2014*; *Yang et al., 2014*).

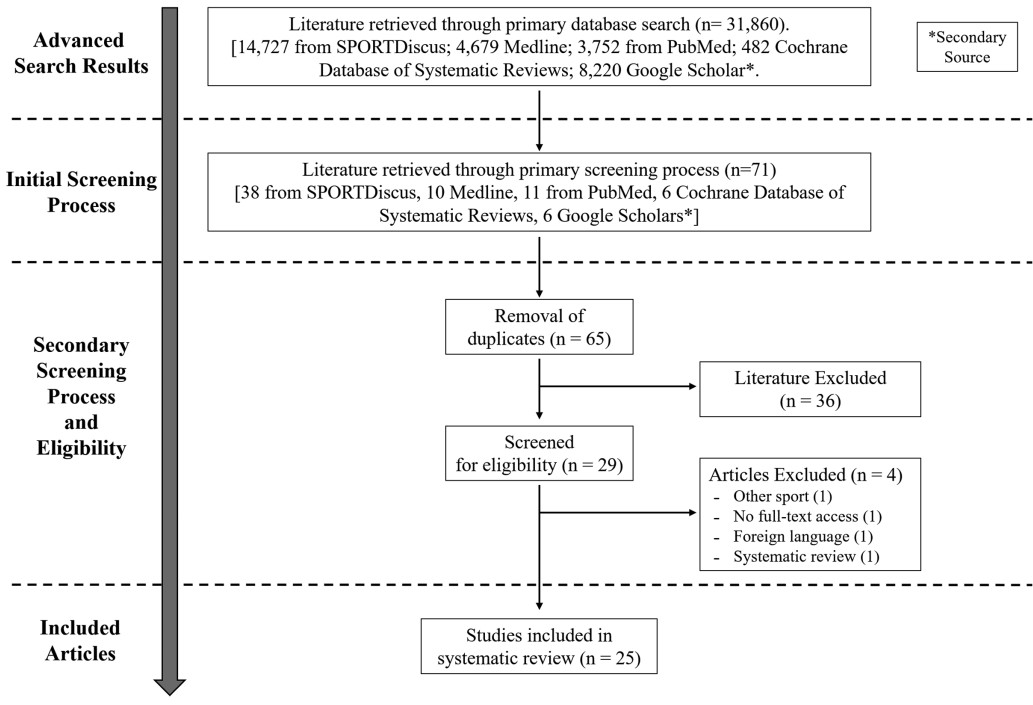

**Figure 1 PRISMA flowchart.** PRISMA flow chart for search and article screening process.

Although many of these observed changes are indicators of fatigue, these kinematic changes may not place additional tissue stress or joint loading on all pitchers, but act as protective mechanisms.

Not all studies note changes in kinematics with fatigue. A controlled lab study directed ten collegiate baseball pitchers to throw 15 pitches per inning in a seven to nine inning simulated indoor game (*Escamilla et al., 2007*). The trunk moved significantly closer to a vertical position, however, pitching biomechanics remained very similar with the onset of fatigue, which was inferred by a significant decrease in ball velocity (*Escamilla et al., 2007*). Two other studies identified a decrease in velocity with accumulation of fatigue (*Crotin et al., 2013*; *Whiteside et al., 2016*), but minimal changes in kinematics.

The literature independently supports that changes in kinematics and/or performance are likely to result from fatigue while pitching. The exact mechanisms of how these changes translate to injury is not entirely clear and the literature has mixed findings on the topic. Therefore, the purpose of this study was to systematically review available literature in an attempt to establish a link between kinematics, performance and injury during the manifestation of muscle fatigue in baseball pitching.

## METHODOLOGY

The review was conducted according to preferred reporting items for systematic reviews and meta-analyses guidelines (*Moher et al., 2009*) (Fig. 1) and the protocol was registered with PROSPERO (*Holmes, Sonne & Birfer, 2018*; ID: CRD42018114194).

## Search strategy

Searches were conducted using four primary online databases, including SPORTDiscus, Medline, PubMed, and Cochrane Database of Systematic Reviews. Google Scholar was a secondary database used to acquire additional literature. Searches were designed with the help of a librarian at Brock University. Articles were discovered using keywords including "fatigue in pitchers" and aspects describing "kinematics" (pitching mechanics, throwing mechanics, valgus elbow torque, joint stiffness, pitching kinematics), "performance" (ball velocity, pitch count, pitch type, spin rate, overhead throwing, fatigue, and performance), and "injury" (elbow pain, shoulder soreness, pitching injuries, fatigue and injuries, muscle fatigue, long-term injuries).

Given the nature of this review, an operational definition of fatigue was required to help guide article selection and inclusion. For this work, fatigue was operationally defined as an exercise induced loss in muscle force generating capacity. This can result in decreased strength (*Enoka & Stuart, 1992*; *Fitts, 1994*; *Vollestad, 1997*), joint stability (*Webster & Nussbaum, 2016*), postural control (*Gribble & Hertel, 2004*; *Bizid et al., 2009*), and altered kinematics (*Apriantono et al., 2006*; *Becker et al., 2017*; *Niederer et al., 2016*).

## Eligibility criteria

Studies were included if: they were published in the English language; they were peer reviewed articles published in journals; they included baseball pitchers of various levels (junior, high school, collegiate, or professional) and were gender specific toward males only. The studies analyzed original research conducted within a laboratory or field-study setting. Longitudinal, epidemiological, retrospective, experimental, conference papers, and crossover laboratory studies were included (i.e., case reports, reviews, editorials, and letters were all excluded). Articles were excluded if the study did not include baseball pitchers, therefore studies examining a population of softball players or other overhead throwing athletes were not included in the review.

## Methodological approach

After an initial screening for our inclusion and exclusion criteria and the removal of duplicates (RB), three reviewers independently assessed the articles by screening abstracts (RB, MS, MH). Next, the full text of each article was obtained and screened against the exclusion criteria. Each reviewer independently indicated the article as either "relevant," "irrelevant" or "possibly relevant." Any disagreements were resolved through a consensus meeting between the three reviewers, such that all remaining articles had complete agreement as "relevant."

## Data extraction

The relevant data was extracted from the included articles and the methods described therein were assessed independently by three reviewers. All extracted results found in the accompanying tables were performed by one reviewer (RB) and two reviewers (MS, MH) cross-referenced and check the extracted data. The included data was extracted based on the category of the article (three categories were established, see below) and

included: first author, year of publication, study setting, number of pitchers included in the study, and outcome measure (e.g., kinematic data, performance data, injury data).

## Assessment of methodological quality

Methodological quality was assessed using quality scores from the Downs and Black Index (*Downs & Black, 1998*). Data from the included articles were extracted from the following categories: authors, year of publication, study purpose, design, population, statistical analysis, and results. A criteria list was compiled, incorporating all of the selected articles used in this systematic review, outlining each study. The Downs and Black assessment was chosen as it has been validated and the original checklist could be modified for the needs of our systematic review (*Downs & Black, 1998*). For this work, 13 items from the original checklist were identified as relevant to this work. Each item was scored as either "yes" = 1, "no" = 0, "unable to determine" = U/0. The scores from all 13 items were totaled to provide the quality score (Table 1).

## Risk of bias assessment

An assessment of risk bias was determined based on the work of *Lopes et al. (2012)* and adapted by *Ceyssens et al. (2019)*. The criteria used by *Lopes et al. (2012)* was adapted for our work, with the scoring system based on the same 10 items: (1) definition of injury clearly described, (2) prospective design that presents incidence or prevalence data, (3) description of level of pitchers (e.g., recreational or professional level), (4) the process of inclusion of athletes in the study was random (i.e., not by convenience) or the data collection was performed with the entire target population; (5) data analysis performed with at least 80% of the athletes included in the study; (6) injury data reported by pitchers; (7) same mode of injury data collection used; (8) injury diagnosis conducted by a medical professional; (9) follow-up period of at least 6 months; (10) incidence or prevalence rates of injury expressed by a ratio that represents both the number of injuries as well as the exposure to pitching (Table 2).

# RESULTS

## Search results

From the extensive database search, eligibility assessment was conducted on 31,860 articles based on their titles, abstracts, and if necessary, full-texts (Fig. 1). Following the initial screening process (based on title and abstract), 71 articles remained after which six duplicate articles were removed, leaving 65 articles. After the completion of the secondary screening process (agreement between all authors), a total of 29 articles were selected to be included for the review. Of the 29 articles, four were removed because: (1) not a baseball study, (2) full-text written in a foreign language, (3) full-text of the study (only abstract viewable) was not accessible to the reviewers and (4) was a systematic review. The remaining 25 articles were included in this review and were binned into three categories based on the studies focus—kinematics ($n = 10$), performance ($n = 13$), and injury ($n = 7$). Note, some articles crossed into more than one category. Given the timeline

**Table 1 Methodological quality assessment.** Methodological quality assessment via Modified Downs and Black quality index.

| Included studies | Modified Downs and Black checklist number | | | | | | | | | | | | | |
|---|---|---|---|---|---|---|---|---|---|---|---|---|---|---|
| | 1 | 2 | 3 | 6 | 7 | 9 | 10 | 11 | 12 | 16 | 18 | 20 | 26 | Total |
| Bradbury & Forman (2012) | 1 | 1 | 1 | 1 | 1 | 0 | 0 | 1 | 1 | U | 1 | 1 | 0 | 9 |
| Chou et al. (2015) | 1 | 1 | 1 | 1 | 1 | 0 | 1 | 0 | 0 | U | 1 | 1 | 0 | 8 |
| Crotin et al. (2013) | 1 | 1 | 1 | 1 | 1 | 0 | 1 | 0 | 0 | U | 1 | 1 | 0 | 8 |
| Crotin et al. (2014) | 1 | 1 | 1 | 1 | 1 | 0 | 1 | 0 | 0 | U | 1 | 1 | 0 | 8 |
| Dale et al. (2007) | 1 | 1 | 1 | 1 | 1 | 0 | 1 | 0 | 0 | U | 1 | 1 | 0 | 8 |
| Erickson et al. (2016) | 1 | 1 | 1 | 1 | 1 | 0 | 1 | 0 | 0 | U | 1 | 1 | 0 | 8 |
| Escamilla et al. (2007) | 1 | 1 | 1 | 1 | 1 | 0 | 1 | 0 | 0 | U | 1 | 1 | 0 | 8 |
| Freeston et al. (2014) | 1 | 1 | 1 | 1 | 1 | 0 | 1 | 0 | 0 | U | 1 | 1 | 0 | 8 |
| Grantham et al. (2014) | 1 | 1 | 0 | 1 | 1 | 0 | 1 | 0 | 0 | U | 1 | 0 | 0 | 6 |
| Keeley, Barber & Oliver (2010) | 1 | 1 | 1 | 1 | 1 | 0 | 0 | 0 | 0 | U | U | 1 | 0 | 6 |
| Keeley et al. (2017) | 1 | 1 | 1 | 1 | 1 | 0 | 1 | 0 | 0 | U | U | 1 | 0 | 7 |
| Lyman et al. (2001) | 1 | 1 | 1 | 1 | 1 | 1 | 1 | 1 | 1 | U | 1 | 1 | 0 | 11 |
| Lyman et al. (2002) | 1 | 1 | 1 | 1 | 1 | 0 | 1 | 1 | 1 | U | 1 | 1 | 0 | 10 |
| Makhni et al. (2014) | 1 | 1 | 1 | 1 | 1 | 0 | 1 | 1 | 1 | U | 1 | 1 | 0 | 10 |
| Mullaney et al. (2005) | 1 | 1 | 1 | 1 | 1 | 0 | 1 | 0 | 0 | U | 1 | 1 | 0 | 8 |
| Murray et al. (2001) | 1 | 1 | 1 | 1 | 1 | 0 | 1 | 0 | 0 | U | 1 | 1 | 0 | 8 |
| Oliver & Plummer (2010) | 1 | 0 | 1 | 0 | 1 | 0 | 1 | 0 | 0 | U | 1 | 0 | 0 | 5 |
| Oliver, Weimar & Henning (2016) | 1 | 0 | 1 | 0 | 1 | 0 | 1 | 0 | 0 | U | 1 | 0 | 0 | 5 |
| Sonne & Keir (2016) | 1 | 1 | 1 | 1 | 1 | 0 | 1 | 0 | 0 | U | 1 | 1 | 0 | 8 |
| Tripp, Yochem & Uhl (2007) | 1 | 1 | 1 | 1 | 1 | 0 | 1 | 0 | 0 | U | 1 | 1 | 0 | 8 |
| Wang et al. (2016) | 1 | 1 | 1 | 1 | 1 | 0 | 1 | 0 | 0 | U | 1 | 1 | 0 | 8 |
| Warren, Szymanski & Landers (2015) | 1 | 1 | 1 | 1 | 1 | 0 | 1 | U | U | U | 1 | 1 | 0 | 8 |
| Whiteside et al. (2016) | 1 | 1 | 1 | 1 | 1 | 0 | 0 | 0 | 0 | U | 1 | 1 | 0 | 7 |
| Yang et al. (2014) | 1 | 1 | 1 | 1 | 1 | 0 | 1 | 1 | 1 | U | 1 | 1 | 0 | 10 |
| Yang et al. (2016) | 1 | 1 | 1 | 1 | 1 | 0 | 1 | 0 | 0 | U | 1 | 1 | 0 | 8 |

**Notes:**

Scoring: "yes" = 1, "no" = 0, "unable to determine" = U (scored as 0).

Criteria: (1) Clear aim/hypothesis, (2) main outcome measures clearly described, (3) patient characteristics clearly described, (6) main findings clearly described, (7) random variability of main outcomes provided, (9) characteristics of patients lost to follow-up described, (10) actual probability values reported, (11) subjects asked to participate representative of entire population, (12) subjects prepared to participate representative of entire population, (16) clear mentioning of data dredging (unplanned analysis), (18) appropriate statistical analysis, (20) valid and reliable outcome measures, (26) patients lost to follow-up taken into account. Note: Only the Downs and Black questions that applied to this work were assessed.

of our search, the earliest, most relevant study included within the review was published in 2001 (*Murray et al., 2001*), whereas the latest came from 2016 (*Yang et al., 2016*).

## Methodological quality and bias assessment

The scores from the modified Downs and Black checklist for each included article is provided in Table 1. Quality scores ranged from five to 11 out of 13 (38–85%). None of our included articles were excluded on the basis of methodological quality. Scores on the modified risk bias scale from *Lopes et al. (2012)* ranged from three to nine out of 10. Of the included articles in this systematic review, two received a score less than five, which can be considered a high risk of bias (*Ceyssens et al., 2019*). The remaining 23 articles

**Table 2  Risk of bias.** Risk of bias assessment related to included studies (modified from *Lopes et al., 2012*).

| Included studies | Criteria | | | | | | | | | | |
|---|---|---|---|---|---|---|---|---|---|---|---|
| | 1 | 2 | 3 | 4 | 5 | 6 | 7 | 8 | 9 | 10 | Total |
| *Bradbury & Forman (2012)* | 0 | 1 | 1 | 1 | 1 | 0 | 0 | 0 | 0 | 0 | 4 |
| *Chou et al. (2015)* | 1 | 1 | 1 | 1 | 1 | 0 | 0 | 0 | 0 | 0 | 5 |
| *Crotin et al. (2013)* | 1 | 1 | 1 | U | 1 | 0 | 0 | 0 | 0 | 0 | 4 |
| *Crotin et al. (2014)* | 0 | 1 | 1 | 1 | 1 | 0 | 0 | 0 | 0 | 0 | 4 |
| *Dale et al. (2007)* | 0 | 1 | 1 | 0 | 1 | 0 | 0 | 0 | 0 | 0 | 3 |
| *Erickson et al. (2016)* | 1 | 1 | 1 | 1 | 1 | 0 | 0 | 0 | 0 | 0 | 5 |
| *Escamilla et al. (2007)* | 1 | 1 | 1 | 0 | 1 | 0 | 0 | 0 | 0 | 0 | 4 |
| *Freeston et al. (2014)* | 0 | 1 | 1 | U | 1 | 0 | 0 | 0 | 0 | 0 | 3 |
| *Grantham et al. (2014)* | 0 | 1 | 0 | 1 | 1 | 0 | 0 | 0 | 0 | 0 | 3 |
| *Keeley, Barber & Oliver (2010)* | 0 | 1 | 1 | 1 | 1 | 0 | 0 | 0 | 0 | 0 | 4 |
| *Keeley et al. (2017)* | 0 | 1 | 1 | 1 | 1 | 0 | 0 | 0 | 0 | 0 | 4 |
| *Lyman et al. (2001)* | 0 | 1 | 1 | U | 1 | 1 | 1 | 0 | 0 | 1 | 6 |
| *Lyman et al. (2002)* | 1 | 1 | 1 | U | 0 | 1 | 1 | 0 | 0 | 1 | 6 |
| *Makhni et al. (2014)* | 1 | 1 | 1 | U | 1 | 1 | 1 | 0 | 0 | 1 | 7 |
| *Mullaney et al. (2005)* | 0 | 1 | 1 | 0 | 1 | 0 | 0 | 0 | 0 | 0 | 3 |
| *Murray et al. (2001)* | 0 | 1 | 1 | 1 | 0 | 0 | 0 | 0 | 0 | 0 | 3 |
| *Oliver & Plummer (2010)* | 0 | 1 | 1 | 0 | 1 | 0 | 0 | 0 | 0 | 0 | 3 |
| *Oliver, Weimar & Henning (2016)* | 0 | 1 | 1 | U | 1 | 0 | 0 | 0 | 0 | 0 | 3 |
| *Sonne & Keir (2016)* | 1 | 1 | 1 | 0 | 1 | 0 | 0 | 0 | 0 | 0 | 4 |
| *Tripp, Yochem & Uhl (2007)* | 0 | 1 | 1 | 0 | 1 | 0 | 0 | 0 | 0 | 0 | 3 |
| *Wang et al. (2016)* | 1 | 1 | 0 | U | 1 | 0 | 0 | 0 | 0 | 0 | 3 |
| *Warren, Szymanski & Landers (2015)* | 0 | 1 | 1 | U | 1 | 0 | 0 | 0 | 0 | 0 | 3 |
| *Whiteside et al. (2016)* | 1 | 1 | 1 | 1 | 1 | 0 | 0 | 0 | 0 | 0 | 5 |
| *Yang et al. (2014)* | 1 | 1 | 1 | 1 | 1 | 1 | 1 | 1 | 0 | 1 | 9 |
| *Yang et al. (2016)* | 0 | 1 | 1 | 1 | 1 | 0 | 0 | 0 | 0 | 0 | 4 |

Note:
Scoring: "low risk of bias" = 1, "high risk of bias" = 0, "unable to determine" = U (scored as 0)

received a low risk of bias score. Item 9 related to a follow-up period of at least 6 months received the lowest score, whereas item 2 relating to design received the highest score. All scores retrieved from the included articles can be found in Table 2.

## Participants and study design

The majority of the included studies assessed athletes at the junior (*Freeston et al., 2014*; *Lyman et al., 2001*, *2002*; *Makhni et al., 2014*; *Oliver, Weimar & Henning, 2016*; *Yang et al., 2014*), high school (*Chou et al., 2015*; *Crotin et al., 2014*; *Erickson et al., 2016*; *Oliver & Plummer, 2010*; *Yang et al., 2014*, *2016*), or collegiate level (*Crotin et al., 2014*; *Dale et al., 2007*; *Escamilla et al., 2007*; *Grantham et al., 2014*; *Keeley, Barber & Oliver, 2010*; *Mullaney et al., 2005*; *Tripp, Yochem & Uhl, 2007*; *Wang et al., 2016*; *Warren, Szymanski & Landers, 2015*) with reported mean age ranges from 8 to 23 years old. Five of the 25 studies included professional baseball players (*Bradbury & Forman, 2012*;

*Crotin et al., 2013*; *Mullaney et al., 2005*; *Murray et al., 2001*; *Sonne & Keir, 2016*; *Whiteside et al., 2016*), but an age range was only reported from one.

The prospective cohorts and longitudinal studies (*Bradbury & Forman, 2012*; *Lyman et al., 2001*, *2002*) examined outcomes of workload across multiple seasons and the decline in performance associated with the onset of fatigue. The epidemiological study used a survey of youth baseball players to investigate arm pain (*Makhni et al., 2014*). A cross-sectional study (*Yang et al., 2014*), included a national survey of youth pitchers who self-reported risk-prone activities. Furthermore, these studies assessed the effects of pitch type, pitching through pain, and rest time on the manifestation of fatigue in baseball pitchers. Retrospective studies included analysis of pitching records in minor league pitchers (*Crotin et al., 2013*) and publicly available data on professional pitchers (*Sonne & Keir, 2016*). Descriptive/controlled laboratory studies analyzed kinematic changes in a pitcher's delivery with the accumulation of fatigue (*Chou et al., 2015*; *Dale et al., 2007*; *Erickson et al., 2016*; *Escamilla et al., 2007*; *Grantham et al., 2014*; *Keeley et al., 2017*; *Mullaney et al., 2005*; *Murray et al., 2001*; *Oliver & Plummer, 2010*; *Oliver, Weimar & Henning, 2016*; *Tripp, Yochem & Uhl, 2007*; *Wang et al., 2016*; *Warren, Szymanski & Landers, 2015*; *Whiteside et al., 2016*). Randomized crossover studies (*Crotin et al., 2014*; *Yang et al., 2016*; *Freeston et al., 2014*) included comparing the effects of throwing exercises and identifying markers for arm fatigue (*Freeston et al., 2014*), kinematic analysis while altering stride length over multiple sessions (*Crotin et al., 2014*) and examining the effects of rest intervals during a simulated game (*Yang et al., 2016*).

## Main findings

Evidence extracted from the 25 studies demonstrates a relationship between fatigue in baseball pitching, and three overarching outcomes: changes in kinematics, a decrease in performance, and an increase in injury risk. Each is discussed in more detail below.

## Kinematic changes

For this review, 10 of the 25 articles primarily assessed the relationship between kinematic changes and fatigue in baseball pitching, mainly in the arm cocking and acceleration phase. These changes are summarized in Table 3. As fatigue accumulated in pitchers, significant differences were seen in maximum shoulder external rotation (*Erickson et al., 2016*; *Mullaney et al., 2005*; *Murray et al., 2001*), knee angle at ball release (*Murray et al., 2001*) and hip-to-shoulder separation (*Erickson et al., 2016*), as well as, other changes highlighted in Table 3.

## Performance changes

Pitching is an activity involving dynamic and high intensity muscle contractions, separated by periods of rest in between pitches (*Dale et al., 2007*). Pitch count, pitch type, ball velocity, and rest time are among the main components that could either positively or negatively impact performance on the mound. Table 4 summarizes the 13 studies that were identified as the fatigue and performance changes category. A study associated decreased stride length with reduced mean pitching heart rate, decreased pitching

**Table 3 Kinematic changes with fatigue.** A summary of selected pitching kinematic changes following various fatigue protocols.

| Study | Sample size | Fatigue protocol | Changes in kinematics due to fatigue | Pre–post fatigue change |
|---|---|---|---|---|
| *Escamilla et al. (2007)* | 10 collegiate pitchers | 15 pitches per inning for seven to nine innings | - Change in trunk flexion during the arm cocking and acceleration phase (from 34° ± 12° to 29° ± 11°) | - 5° change in trunk flexion |
| *Murray et al. (2001)* | Seven major league pitchers | No protocol, collection during games | - Decreased maximum shoulder external rotation (181° in the first inning to 172° in the last)<br>- Decreased knee angle at ball release (140° in first inning to 132° in the last) | - 9° change in maximum external rotation<br>- 8° change in knee angle |
| *Erickson et al. (2016)* | 28 adolescent pitchers | Warmup followed by 15 pitches per inning for six innings | - Hip-to-shoulder separation decreased as pitch count increased (from 90% ± 40% at pitch 15 to 40% ± 50% at pitch 90; $p < 0.001$)<br>- Knee flexion increased with pitch number (from 49° ±15° to 53° ± 15°, $p = 0.008$)<br>- Increased shoulder external rotation and total range of motion post pitching<br>- Lower half muscles fatigued before changes in upper extremity kinematics occurred | - 50% change in hip-to-shoulder separation<br>- 4° change in knee flexion |
| *Mullaney et al. (2005)* | 13 university pitchers | 99 pitches over seven innings | - Postgame results showed selective fatigue of 15% in shoulder flexion ($p = 0.02$); 18% in shoulder internal rotation ($p = 0.03$); and 11% in shoulder adduction ($p = 0.01$) | - N/A |
| *Keeley, Barber & Oliver, 2010* | 10 collegiate pitchers | Five pitches for strikes followed by two kg ball throws until maximum perceived fatigue | - Changes in lateral pelvis tilt at maximum external rotation (from −10.8° ± 11.8° to −14.8° ± 11.3°) and ball release (from −3.36° ± 5.24° to −6.82° ± 3.87°) between non-fatigued and fatigued conditions ($p < 0.05$) (negative represents tilt to left) | - 4° change in lateral pelvis tilt at maximum external rotation<br>- 3.5° change in lateral pelvis tilt at ball release |

*(Continued)*

| Study | Sample size | Fatigue protocol | Changes in kinematics due to fatigue | Pre-post fatigue change |
|---|---|---|---|---|
| *Chou et al. (2015)* | 16 high school pitchers | 10 maximum effort fastball warmups, 100 pitches, 10 pitches post throwing session | - Increased knee flexion (from $53.6° \pm 21.5°$ to $56.1° \pm 22.2°$; $p = 0.01$) and trunk flexion (from $21.4° \pm 5.4°$ to $24.2° \pm 6.6°$; $p = 0.01$) at instant of ball release<br>- Shoulder horizontal abduction decreased at the instant of front foot strike (from $21.1° \pm 11.0°$ to $18.7° \pm 10.1°$; $p = 0.01$)<br>- Maximum forearm pronation decreased during the acceleration phase (from $27.4° \pm 12.3°$ to $22.4° \pm 10.6°$; $p = 0.01$)<br>- Elbow valgus ($8.8° \pm 3.3°$ to $6.9° \pm 3.9°$; $p = 0.01$) and forearm pronation ($24.9° \pm 11.2°$ to $20.1° \pm 8.9°$; $p = 0.01$) decreased at ball release as fatigue accumulated | - $2.5°$ change in knee flexion at ball release<br>- $2.8°$ change in trunk flexion at ball release<br>- $2.4°$ change in horizontal abduction at front foot strike<br>- $5°$ change in forearm pronation during acceleration<br>- $1.9°$ change in elbow valgus angle at ball release<br>- $4.8°$ change in forearm pronation at ball release |
| *Oliver & Plummer (2010)* | 14 high school pitchers | Five pitches for strikes followed by two kg ball throws until maximum perceived fatigue | - Kinematic data was collected, but results were not shown within the study's results | - N/A |
| *Oliver, Weimar & Henning (2016)* | 23 youth pitchers | 75 pitch limit | - Kinematic data was collected, but results were not shown within the study's results | - N/A |
| *Tripp, Yochem & Uhl (2007)* | 16 collegiate pitchers | Three to five warmup pitches, throwing every 5 s until maximum perceived fatigue | - Arm cocked position changed from 12.4 mm pre-fatigue to 24.1 mm post-fatigue (decreased acuity)<br>- Ball release position changed from 20.8 mm pre-fatigue to 41.7 mm post-fatigue (decreased acuity) | - 11.7 mm change in joint position sense in arm cocked position<br>- 20.9 mm change in joint position sense at ball release |
| *Grantham et al. (2014)* | 11 collegiate pitchers | No protocol, collection during games | - Increased hip flexion at hand separation ($p = 0.022$)<br>- Increased hip flexion ($p = 0.002$) and shoulder lateral tilt ($p = 0.048$) at maximum external rotation was observed in innings in which the pitcher threw over 15 pitches | - N/A |
**Table 4 Performance changes with fatigue.** Studies examining velocity prior to, and post-fatigue protocol. Absolute velocity, pre and post fatigue, as well as the relative change in velocity. Additionally, studies related to throwing accuracy listed.

| Study | Sample size | Fatigue protocol | Velocity pre-fatigue | Velocity post-fatigue | Relative change (% velocity decrease from pre-fatigue) | Throwing accuracy/other |
|---|---|---|---|---|---|---|
| Dale et al. (2007) | A total of 10 collegiate pitchers | A total of 60 maximum effort pitches, 15 each inning | 82.5 ± 1.3 mph | 81.5 ± 0.9 mph | −1.2 % | N/A |
| Crotin et al. (2014) | A total of 19 collegiate/high school pitchers | Warmup. A total of 80 pitches (15 seconds between pitches, 9 min between innings) | Over-stride: 81.6 ± 5.4 mph Under-stride: 80.3 ± 5.0 mph | Over-stride: 79.8 ± 5.4 mph Under-stride: 79.8 ± 5.0 mph | Over-stride: −2.2% Under-stride: −0.6% | N/A |
| Keeley, Barber & Oliver (2010) | A total of 10 collegiate pitchers | Five pitches for strikes. two kg ball throws until maximum perceived fatigue | 75.0 mph | 72.0 mph | −4.0% | N/A |
| Murray et al. (2001) | Seven major league pitchers | No protocol, collection during season | 90.0 mph | 85.0 mph | −5.6% | N/A |
| Erickson et al. (2016) | A total of 28 male pitchers | Warmup. A total of 15 pitches per inning for six innings | 73.0 ± 5.0 mph | 71.0 ± 6.0 mph | −2.7% | N/A |
| Escamilla et al. (2007) | A total of 10 collegiate pitchers | A total of 15 pitches per inning for seven to nine innings | 77.6 ± 4.0 mph | 75.4 ± 3.4 mph | −2.8% | N/A |
| Whiteside et al. (2016) | A total of 129 MLB pitchers | No protocol, collection during season | N/A | N/A | N/A | Percentage of hard-thrown pitches decreased as game progressed. Largest decrease in ball speed between 1st and 7th inning (velocity not provided) |
| Wang et al. (2016) | A total of 15 pitchers | Six maximum effort fastballs before fatigue protocol. 8–12 reps, three sets, wrist ulnar deviation and flexion with dumbbell. Within 1 min of completion, pitcher threw six maximum effort fastballs | N/A | N/A | N/A | Strike percentage changed from 70.1 ± 17.8% pre-fatigue to 49.3 ± 17.2% post-fatigue |

(Continued)

| Table 4 (continued). | | | | | | |
|---|---|---|---|---|---|---|
| Study | Sample size | Fatigue protocol | Velocity pre-fatigue | Velocity post-fatigue | Relative change (% velocity decrease from pre-fatigue) | Throwing accuracy/other |
| Crotin et al. (2013) | A total of 12 minor league pitchers | No protocol, collection during season | N/A | N/A | N/A | Home run rate increased with each pitch |
| Bradbury & Forman (2012) | A total of 1,058 MLB pitchers | No protocol, collection during seasons | N/A | N/A | N/A | With each pitch in preceding game, 5th game and 10th game, the pitcher's Earned Run Average increased by 0.007, 0.014, and 0.022, respectively |
| Yang et al. (2016) | Seven intercollegiate pitchers | A total of 15 pitches per inning for seven innings | N/A | N/A | N/A | Both throwing accuracy and velocity significantly decreased below baseline following the 4th inning in the 8-s ($p = 0.05$) and 12-s ($p = 0.05$) trials |
| Keeley et al., 2017 | A total of 14 youth pitchers | A total of 88 pitch simulated game | N/A | N/A | N/A | Total and first pitch strike percentage decreased at "moderate" perceived fatigue levels (52.4% and 49.8%) and further at "severely" fatigued (45.3% and 40.0%) |

intensity, improved recovery capacity, and lowered salivary cortisol from baseline (Crotin et al., 2013). Accumulation of fatigue also accompanied decreases in throwing accuracy (Wang et al., 2016; Yang et al., 2016) and future performance (Bradbury & Forman, 2012).

## Fatigue linked to pain and injury

Pitching with discomfort, pitching through pain, and pitching with tiredness are three primary cues or precursors to injury. Table 5 summarizes the seven articles that were

**Table 5 Injury and fatigue.** A summary of findings related to pain and injury resulting from pitching.

| Study | Sample size | Fatigue protocol | Data collection process | Findings |
|---|---|---|---|---|
| *Lyman et al. (2001)* | 298 youth pitchers (aged 9–12 years) | No protocol, collection during season | Conducted over the span of two seasons, pitchers were interviewed via telephone after each game pitched | - Elbow pain was reported in 26% of pitchers, while shoulder pain was reported in 32% of pitchers<br>- Risk factors associated with elbow and shoulder pain included decreased self-satisfaction, increased pitch count, and in-game arm fatigue<br>- Increased age, weight, and lifting weights during the season linked to increased elbow pain |
| *Lyman et al. (2002)* | 476 youth pitchers (aged 9–14 years) | No protocol, collection during season | Questionnaires were assigned to pitchers before and after the season. Interviews were conducted during the season after each game | - Curveballs were associated with a 52% increased risk of shoulder pain, while the slider was associated with an 86% increased risk of elbow pain<br>- 28% of pitchers reported elbow pain and 35% of pitchers reported shoulder pain at least once during the season<br>- Elbow and shoulder pain increased significantly with pitch count |
| *Freeston et al. (2014)* | 13 elite pitchers (aged 19.6 ± 2.6 years) | Two test days (minimum of 7 days apart), 5–10 min of moderate intensity running, 5–10 min of stretching, 10–15 min of throwing, throwing or running program | A throwing protocol was assigned to subjects on the first day, a running protocol was assigned on the second day | - Significant increase in velocity following the throwing program (3.5 ± 0.7 vs. 1.4 ± 0.5 km/h, respectively; $p \leq 0.05$)<br>- Throwing velocity, a sign of general fatigue, whereas throwing accuracy and arm soreness are indicators of arm fatigue |
| *Yang et al. (2014)* | 754 youth pitchers (aged 9–18 years) | No protocol, collection during season | A national survey was conducted | - 69.2% of pitchers reported pitching through arm tiredness multiple times throughout the season<br>- 37.9% of pitchers reported multiple incidences of arm pain throughout the season |
| *Makhni et al. (2014)* | 203 youth pitchers (aged 8–18 years) | No protocol, collection during season | Epidemiological study. Survey. | - 23% of pitchers reported prior overuse injury<br>- 30% of players reported arm pain at decreased level of satisfaction<br>- 46% of players were told on at least one occasion to pitch through arm pain |

(Continued)

| Study | Sample size | Fatigue protocol | Data collection process | Findings |
|---|---|---|---|---|
| *Sonne & Keir (2016)* | 73 pitchers | No protocol, collection during season | Retrospective Study. Pitching data retrieved from a public database. | - Reduced effectiveness of the flexor-pronator mass reduces joint rotational stiffness, which in turn increases the strain on the UCL during pitching, therefore increasing risk of injury |
| *Warren, Szymanski & Landers (2015)* | 21 collegiate pitchers (aged 20.4 ± 1.4 years) | Three simulated, five-inning games (Max 70 pitches per game) | Evaluated the effects of three recovery protocols on range of motion, heart rate, rating of perceived exertion, and blood lactate | - Study looked into injuries in pitching, but did not comment on any related findings |

included in the fatigue and pain/injury/discomfort category. A prospective cohort study found that elbow pain increased significantly with slider usage, while shoulder pain increased significantly with curveball usage (*Lyman et al., 2002*). Furthermore, multiple studies reported elbow and shoulder tiredness and pain throughout a season (*Lyman et al., 2001*, *2002*; *Yang et al., 2014*), which were identified to be associated with decreased self-satisfaction and elevated pitch count (*Lyman et al., 2001*; *Makhni et al., 2014*). A randomized crossover trial compared ball velocity post-throwing and post-running programs, and while each program led to a significant increase in arm soreness, there was a larger increase in ball velocity following the throwing program (*Freeston et al., 2014*).

## DISCUSSION

Following an extensive screening process, evidence was drawn from longitudinal, retrospective, epidemiological, experimental, and laboratory studies. To our knowledge, this is the first review to extract evidence from available literature and systematically identify a relationship between kinematics, performance, and injury during the manifestation of muscle fatigue in baseball pitchers. The main findings of this systematic review identified a co-dependence between changes in kinematics to delay a decrease in performance, which could result in an increased risk of musculoskeletal injury (Fig. 2).

This work suggests that changes in performance likely decay at a lesser rate than changes in kinematics, suggesting that modifications to a pitcher's kinematics are made to limit the decrements of fatigue. The compromise for maintaining performance is the adaptation of mechanics which could increase the risk of musculoskeletal injury, particularly with respect to the elbow and shoulder joints. Many of the studies examined in this review also included measures of muscle activity via electromyography (EMG). While EMG can be loosely used as a surrogate indicator of force (*Roberts & Gabaldon, 2008*) it is also an indicator of muscle fatigue (typically represented by increases in EMG amplitude and decreases in mean and median power frequency). Surface EMG is the

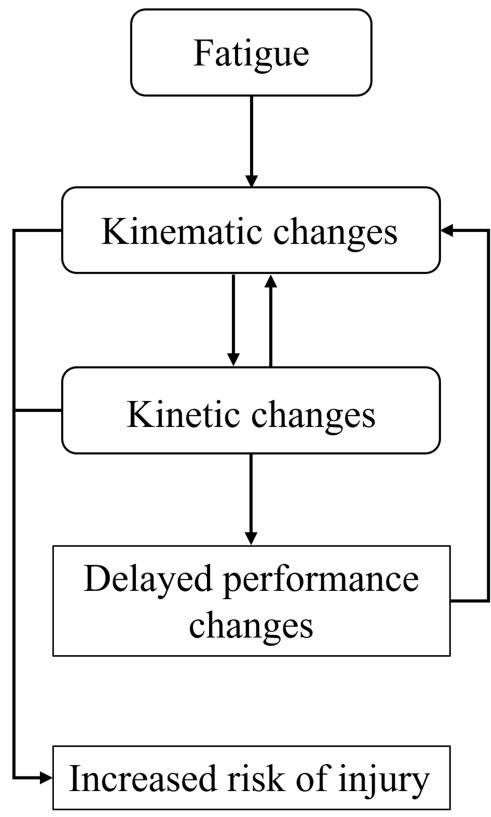

**Figure 2 Theoretical framework summary.** Theoretical framework representing the relationship between fatigue and our three identified outcomes.

recording of electrical signals transmitted from the brain to the neuromuscular junction, resulting in the contraction of the muscle, and the production of a force at the end effector (*Dowling, 1997*; *De Luca, 1997*). While consideration must be made surrounding factors of movement, fatigue, skin impedance, and a variety of other physiological and environmental factors related to the interpretation of EMG, the use of surface EMG in sports is a non-invasive method of predicting internal forces as a result of human movement (*Clarys et al., 2010*). There are challenges with inferring the results from EMG during pitching, particularly with changes in kinematics. During such a dynamic activity, the EMG signal must be interpreted with caution of the potential limitations, such as the electrode position over the muscle belly changing during rapid, ballistic movements—particularly when a change in kinematics results in different limb position identified during a specific position during the throwing motion. As a result, it is difficult to identify if these studies have seen an increase in EMG activity as a result of increased motor unit recruitment due to fatigue, a requirement for increased muscle force, or due to the nature of EMG limitations during dynamic movements. Muscle fatigue reduces the force generating capacity of the elbow and shoulder muscles, thus compromising the potential to maintain joint stability, and thus, injury risk increases. This work summarizes the changes in kinematics, performance and injury risk as a result of fatigue, while acknowledging the difficulty of direct fatigue measures.
## Kinematic changes

Changes in kinematics are a primary outcome of pitching-induced muscle fatigue (Table 3). With the accumulation of muscle fatigue, trunk flexion is altered during the arm cocking and acceleration phase (from $34 \pm 12°$ to $29 \pm 11°$), as well as observed changes in shoulder and elbow kinetics, all of which can increase the risk of injury in pitchers (Escamilla et al., 2007). After the implementation of fatigue protocols in laboratory settings, recovery time varied for the scapulothoracic, wrist, elbow, and glenohumeral joints wherein the latter took an extended amount of time to recover (Tripp, Yochem & Uhl, 2007). Moreover, these kinematic changes may partly be explained by decreases in triceps muscle activity, and an increase in activity for the biceps and deltoids as pitchers reached fatigue (Oliver & Plummer, 2010). Other noted signs of kinematic changes with the onset of fatigue are small, yet potentially significant, increases in knee flexion (from $53.6 \pm 21.5°$ to $56.1 \pm 22.2°$) and forward trunk tilt at the instant of ball release (Chou et al., 2015). With the accumulation of innings, significant decreases in maximal shoulder external rotation, knee angle at ball release, and elbow flexion have been observed as a pitcher becomes fatigued (Grantham et al., 2014; Murray et al., 2001). A controlled laboratory study of a simulated baseball game suggested that collegiate pitchers maintain their kinematics throughout the duration of a game, despite fatigue (Escamilla et al., 2007). Escamilla et al. (2007) found that trunk kinematics changed, yet other kinematic and kinetic variables were unaffected. The authors did note that a more rigorous fatigue protocol may provide additional insight. However, even slight deviations from a pitcher's optimal kinematic patterns, due to the manifestation of fatigue, or pitching in a fatigued state for a longer period of time, may increase the risk of injury in pitchers (Chalmers et al., 2017; Escamilla et al., 2007). These kinematic changes may serve as a precursor to changes in performance, and eventual injury risk.

## Kinetic changes

Changes in kinematics would in turn lead to changes in kinetics, as the body attempts to compensate for the negative effects instigated by the accumulation of muscular fatigue. After videotaping seven major league pitchers for multiple innings, changes in kinematics led to decreases in maximum distraction forces at both the shoulder (from 97% of body weight in the first inning to 88% of body weight in the last) and elbow (from 85% of body weight in the first and 72% of body weight in the last), and horizontal adduction torque at ball release and maximum horizontal abduction torque (from 5% and 11% of body weight, respectively, in the first inning, down to 4% and 8% of body weight in the last inning), but ball velocity did ultimately decrease by five mph (Murray et al., 2001). A study assessing the contribution of forearm flexor muscles during pitching noted that peak flexor carpi ulnaris muscle activity was significantly greater during fastballs post a fatigue protocol during the acceleration phase of the pitching motion to provide more stability for the wrist joint (confidence interval [0.49–2.05]; $p = 0.02$; $d = 1.27$) (Wang et al., 2016). A separate study examined the importance of stabilizing the wrist, identifying peak flexor carpi ulnaris muscle activity to be significantly greater for the fastball post-fatigue protocol during the acceleration phase of the pitch. Since both the

flexor carpi ulnaris and radialis muscles attach to the medial side of the elbow, the accumulation of fatigue may negatively affect the ulnar nerve (*Wang et al., 2016*). In addition to the noted work that highlighted wrist flexor muscle fatigue, a retrospective analysis identified that the greatest muscle fatigue during a game occurs in the extensor carpi radialis (*Sonne & Keir, 2016*). However, the authors also acknowledged that elevated fatigue in the wrist flexors (predominantly flexor digitorum superficialis and pronator teres), are noteworthy, given their large contribution to stabilize the elbow and counter valgus torque. With an increase in muscle fatigue, there will be a reduction in overall elbow joint stiffness, which can ultimately increase the likelihood of an UCL injury (*Sonne & Keir, 2016*). While this review aimed to evaluate directly measurable outcomes in kinematics and performance, it is acknowledged that a more in-depth evaluation of kinetics (which are typically more difficult to quantify than kinematics and even more challenging to quantify in a fatigued state), in relation to changes in kinematics, is a valuable next step.

## Performance changes

Numerous studies have noted that changes in kinematics during a baseball game (due to fatigue) can lead to a significant decrease in ball velocity and therefore can impact performance (*Whiteside et al., 2016*). Furthermore, retrospective analysis studies have not only confirmed a decrease in ball velocity due to fatigue, but have also provided evidence that with each pitch thrown in the preceding game, there is a significant increase in earned run average and the home run rate with each pitch thrown (*Crotin et al., 2013*; *Bradbury & Forman, 2012*). Alternatively, *Keeley et al. (2017)* identified the effect of fatigue on throwing accuracy (Table 4). After a sample of 14 youth pitchers were recruited for the study, results showed that both total strike percentage and first pitch strike percentage decreased at a perceived fatigue level of "moderate" (52.4% and 49.8%) and further at the "severely" fatigued (45.3% and 40.0%) perceived level (*Keeley et al., 2017*).

A separate study showed the proportion of hard-thrown (fastball type) pitches in the seventh inning decreased compared to the first inning and pitchers threw more off-speed and breaking pitches later in games (*Whiteside et al., 2016*). These findings are supported further by *Whiteside et al. (2016)*, which showed that pitchers tend to compensate for fatigue in later innings, by throwing fewer hard pitches and more offspeed and breaking pitches. Pitchers with a wider repertoire of pitches see more changes in muscle activity patterns with different pitch types, therefore lowering risk of elbow injury due to overuse and fatigue (*Whiteside et al., 2016*). Alternating task types and increasing the variability of biomechanical exposures to the human body has been hypothesized as an injury prevention intervention in ergonomics research (*Srinivasan & Mathiassen, 2012*), and may serve as an explanation for the protective effect of having more pitch types. Nevertheless, these findings are important for attempting to not only quantify effects of fatigue on performance, but how changes in kinematics due to the manifestation of fatigue can influence overall performance.

## Fatigue linked to injury

Fatigue has been related to changes in full body kinematics, in particular to the elbow and shoulder, which, when combined with poor recovery, can decrease tissue tolerance and increase the risk of injury in baseball pitchers (*Oliver & Plummer, 2010*). Current literature has derived evidence that the accumulation of fatigue during pitching can lead to elbow and shoulder discomfort, pain, and/or injury. Risk of injury is known to increase with age and level of competition, because of microtrauma, insufficient rest, and overuse associated with continuous throwing. In fact, one longitudinal study reported 23% of little league players have identified prior overuse injuries (*Lyman et al., 2001*). Risk of injury has also been linked to pitch type, as multiple studies have shown the use of curveballs and sliders contributes to an increased risk of shoulder and elbow injuries (*Lyman et al., 2001*; *Yang et al., 2014*). Moreover, pitchers have reported being encouraged to pitch through arm fatigue and pain/discomfort on several occasions during the span of a season, which increases the likelihood of an eventual injury (*Lyman et al., 2001*). A limitation of available studies is the prevalence of bias due to the use of self-report criteria. Interviews were conducted and questionnaires were asked to be completed independently, which led to subjective responses amongst the population. In experimental and laboratory studies, the Borg scale was incorporated to allow the subjects to subjectively rate their perceived level of exertion, while follow-up questionnaires and telephone interview were conducted following competition in on-field case studies (*Lyman et al., 2001*, *2002*; *Makhni et al., 2014*; *Yang et al., 2014*). While survey and subjective fatigue measures must be met with caution when distinguishing a definitive link between fatigue and injury risk in pitchers, our work suggests evidence of such. This is in agreement with a review by *Bruce & Andrews (2014)* who identified fatigue as a mechanism for injury of the UCL, as well as work by *Fleisig et al. (2009)* who reported that pitchers who threw with regular arm fatigue were 36 times more likely to sustain an injury.

It must be noted that pain and discomfort does not always suggest injury and biomechanical damage to a tissue does not always suggest that an individual will experience pain. Our work highlights that fatigue may increase the risk of injury, but is not necessarily a direct link to injury for all individuals. The work by *Lyman et al. (2001*, *2002*) are some of the only studies to establish a link between fatigue and pain, with inferences to injury. Our search criteria mostly included work from junior and collegiate athletes, rather than professional athletes (although some papers did include this data) and this should be noted as a confounding variable when linking fatigue to injury. Clearly, more work is needed in the area of fatigue as an injury mechanism during pitching. Despite the differences in biomechanical reporting of kinematic and kinetic data during pitching studies, a meta-analysis would help determine true effects of kinematic and kinetic changes due to fatigue and would aid in the establishment of a proper fatigue-injury paradigm.

## CONCLUSIONS

This review has extracted evidence from longitudinal, retrospective, epidemiological, experimental and laboratory studies, deducing a co-dependence between changes in

kinematics and a decrease in performance, stemming from the manifestation of fatigue. This can indirectly suggest, despite the few longitudinal studies that have directly investigated it, that there is a relationship between fatigue and increased risk of injury. To our knowledge, this is the first review of its kind to holistically explore and summarize the literature on the multi-faceted impact of muscle fatigue as it relates to pitching. With a proof of concept now established, the prevention of the negative outcomes of fatigue must be the focus of future research as it is imperative to protect pitchers at all age levels. This work provides insight into how the physical demands of pitching can influence kinematics, performance and potential injury. Specific markers have been identified (kinematic and performance based) that might suggest overloading and additional recovery would be required.

### Funding
The authors received no funding for this work. The funders had no role in study design, data collection and analysis, decision to publish, or preparation of the manuscript.

### Grant Disclosures
The following grant information was disclosed by the authors:
The authors received no funding for this work.

### Competing Interests
Michael Holmes is an Academic Editor for PeerJ. Michael Sonne is the acting research director for the baseball development group, responsible for coordinating research projects between the BDG facility, its members, and universities.

### Author Contributions
- Richard Birfer conceived and designed the experiments, performed the experiments, analyzed the data, contributed reagents/materials/analysis tools, prepared figures and/or tables, authored or reviewed drafts of the paper, approved the final draft.
- Michael WL Sonne conceived and designed the experiments, performed the experiments, analyzed the data, contributed reagents/materials/analysis tools, prepared figures and/or tables, authored or reviewed drafts of the paper, approved the final draft.
- Michael WR Holmes conceived and designed the experiments, performed the experiments, analyzed the data, contributed reagents/materials/analysis tools, prepared figures and/or tables, authored or reviewed drafts of the paper, approved the final draft.

### Data Availability
No raw data were generated that is not already found in the review.

### Supplemental Information
Supplemental information for this article can be found online at http://dx.doi.org/10.7717/peerj.7390#supplemental-information.

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
