# Peer review of "Manifestations of muscle fatigue in baseball pitchers: a systematic review"

_PeerJ, doi:10.7717/peerj.7390_

## Round 0.1 · original submission · Major Revisions

Thank you for considering PeerJ for your research. Your paper has now been reviewed and it was considered to be an interesting and worthy topic. Having said that, there are several major concerns with the paper that need to be addressed. The main point that needs to be considered is the method of the review to ensure it is truly a systematic, not narrative, review. Please find with this email the comments of the reviewers. In revising your paper, please pay careful consideration to the comments and make changes accordingly. Pay particular attention to the methods used for your review as both reviewers raised concerns with this.

Reviewer 1 ·

Basic reporting

Reporting can be improved to make sure this review meets the standard expected of systematic reviews, based on the PRISMA guidelines, i.e. introduction should be more concise, methods and results should include risk of bias assessment, justify weighted average approach to data synthesis, ensure flow chart numbers are correct, provide a table with study characteristics for all included studies, avoid duplication of information in tables and text, ensure information in the discussion is consistent with the results. Please see General Comments section for specific details and strategies to address these issues.

Experimental design

The major concerns with the design are the lack of risk of bias assessment and inclusion of non-pitcher based studies (i.e. swimmers, volleyball, tennis players). These other sports were included in the search methodology but then they are not mentioned in the results and discussion, so it is unclear how they have been integrated. I strongly suggest removing these studies from the eligibility criteria and results. Please see General Comments section for specific details and strategies to address these issues.

Validity of the findings

The authors offer interesting insights based on the findings of their review, but greater care needs to be taken to clearly identify when there is and is not strong evidence to support their claims. It is important to know when evidence is weak or non-existent because it provides direction for future research. Please see General Comments section for specific details and strategies to address these issues.

Additional comments

The authors have chosen an interesting topic to review and I congratulate them on the effort involved in screening 31,860 abstracts. I think this manuscript needs to be improved for publication and I have provided the following specific feedback for the authors that I hope they will find useful.

ABSTRACT
“is a primary reason for injury in baseball pitchers” – too strong a conclusion based on the evidence you have collated
“Findings of interest were not limited to baseball pitchers” – this does not make sense, given the aim and the conclusion, the review should be limited to pitchers.
INTRODUCTION
Line #64: “Fatigue can lead to performance decrements and the corresponding cumulative loading…” – I don’t think the “performance decrements” component fits in this sentence, I suggest just focussing on cumulative loading or better explain how the performance decrements are associated with the cumulative loading.
Line #68-74: restructure this information so that it flows better from the preceding sentences. This paragraph is long, so it will be easy to lose the reader if you jump around too much. These final 2 sentences need to be clearer and more concise to improve readability.
I don’t believe Table 1 is needed. Tables are not expected in the introduction of a journal article.
Line #79: “factors involving the brain and central nervous system” – this is too vague. Please be more specific or just say “due to central and peripheral neural mechanisms”.
Line #78-109: Too much detail here about each individual study. Please make this paragraph more concise so that the important information is better highlighted to the reader e.g. studies showing similar findings can be described together. Introduction sections of literature reviews do not need to be long.
Line #92-93: Good point.
METHODS
You should provide an operational definition of fatigue in the methods section. In the intro you define fatigue as “as a decrease in force generating capacity” but throughout the review you indirectly infer fatigue from high pitch counts, lack of rest, etc.
Line #122: There is no risk of bias assessment, so the review is not consistent with PRISMA. Please justify why this was not done and revise the statement about PRISMA guidelines accordingly.
Line #139-140: “not limited to baseball (swimmers, volleyball, and tennis players)” – I strongly disagree with the use of other sports. Your review objective and search terms are baseball specific, so it does not make sense for your included studies to not be baseball specific. Reviews need to be specific otherwise it is too difficult to draw strong conclusions. Additionally, there is no mention of non-baseball specific findings in the results so it is not clear how they have been incorporated. You should remove the non-baseball studies from the review.
Line #147: separate the data extraction and study selection paragraphs as per the PRISMA checklist. Please also make it clear that data extraction was done by one or multiple reviewers. It is unclear from the current wording.
RESULTS
Line #165-166: 74 minus 7 does not equal 65. Please clarify the numbers here.
Figure #1: several values do not add up. i.e. In the ‘Initial Screening Process” section, 38+10+11+6+6 = 71 not 72; in the ‘Secondary Screening…” section it is not clear how you go from “abstracts screened for eligibility (n=29)” to “full-text articles screened for eligibility (n=9)” to “Articles included (n=26)”. It would be clearer if the side arrows branched off of the descending arrows so that it is clearer which step they came before and after.
In general, these errors suggest that you should undertake a thorough cross-checking of you values to make sure there are no other errors/discrepancies in the final publication.
Line #169: not sure what is meant by “full-text was unavailable for use”. Please make this clearer.
Participants and Study Designs: this information should be in a table summarising the general characteristics of the included studies. The table should have (at a minimum) sample size, participant info, study design and duration (if relevant), outcome measures, key findings. This information is needed to be consistent with PRISMA.
Line #187-189: “The study designs varied within the selected literature, among which included four epidemiological studies, three retrospective analyses studies, two included crossover designs, and one longitudinal study.” – this begs the question, what was the study design of the other 16 included studies?
Line #192: “tiredness” – vague term, please clarify/define further.
Line #208-233: Much of this information is straight repetition of findings and values from Table 2. This paragraph should be condensed to avoid duplication of information from the table. i.e. synthesise the findings and make statements relating to the overall trends i.e. “There were significant changes in kinematics at the shoulder in X studies, elbow in X studies, hip in X studies … The variables that were most commonly affected were X, Y, and Z.”
Table 2: Include a description of the fatigue protocol used in each study. Switch the table to landscape orientation to facilitate this. Describe the age/level of pitchers for each study. Report key null findings (if relevant) as well as significant findings, otherwise it’s a biased representation.
Line #238-263: same feedback from Line #208-233/Table 2 applies here with regards to Table 3.
Table 3: you present weight averages here (i.e. a form of meta-analysis) but this is not described in the methods or highlight in the results texts.
Line #270-274: This is too descriptive to be in the results section. Include in the introduction/discussion if you must.
Line #274-298: Summarise these studies in a table and then succinctly highlight the key findings in-text to improve readability. You should present the findings based on the hierarchy of evidence i.e. RCT first (high quality evidence), survey last (low quality evidence). Talk about the results of studies that used similar study designs together i.e. currently you jump back and forth between longitudinal studies and survey studies and RCTs.
DISCUSSION
#304-306/Figure 2: you don’t have strong enough evidence to claim this. You have studies showing fatigue effects biomechanics and fatigue effects performance but nothing demonstrating that biomechanical changes caused by fatigue are causing performance changes or injury risk (if this information exists it does not appear to be included in your review). You can still propose your theoretical framework but need to highlight the parts that are based on evidence and the parts that are not. This is not necessarily a bad thing because it highlights areas of future research (i.e. the parts of your framework that lack evidence). Please revise the discussion section accordingly.
Line #310: This explanation does not fit with your results which suggest “results showed not only a significant reduction in ball velocity with fatigue,” (Line #254).
Line #311: Not sure what you mean by “command” here. Please make this clearer?
Line #313: these subsequent sentences don’t fit with the start of the paragraph. It’s not clear why you are even bringing up EMG at this point because you haven’t mention it once in your review up until this point. I suggest removing (or condensing and moving to later in the discussion) and elaborating more on the opening concept of this paragraph, which was relationship between fatigue, biomechanics and injury.
Line #337-340: as above, why are you now bringing up EMG.
Line #350-352: need to be careful with matching references to statements like this. The study you have referenced did not investigate injuries so doesn’t support your statement.
Line #356-363: again, this does not support your statement in line #310 that biomechanics change to maintain performance, i.e. here biomechanics have led to reduced force.
Line #363-370: This section is meant to be about kinetics (i.e. based on the subheading) but here you are talking about muscle activity. You’ve only referred to one study about kinetics.
Line #373-374: again, this does not support your interpretation at line #310 about biomechanical changes helping to maintain performance.
Table 4 – this appears to be results copied from 1-study that has been published previously. It would be more appropriate to summarise the findings of this study in-text and include a reference. This table should be deleted. It does not add anything to the interpretation.
Line #401 – this is probably the key paragraph (i.e. injury seems to be a big focus of your overarching review objectives) so you need to make it clearer what the current interpretation of the literature is, i.e. do we have any studies that actually link fatigue with increase injury risk? They must be highlighted first at the beginning of the paragraph. The weaker, survey-based studies can be mentioned later because they provide a low level of evidence that is not convincing in this case.
Line #426: “This can lead to an increased risk of injury in baseball pitchers.” – you have not provided strong enough evidence to make this claim, i.e. please soften your wording e.g. indirect evidence suggests there is a relationship, but there have been few longitudinal studies that have directly investigated it, and fatigue is often inferred from pitch counts, etc.

·

Basic reporting

This manuscript is well written and provides significant background and context on the relationship between fatigue, injury and performance in baseball players. I have provided specific comments on the basic reporting provided within the manuscript with suggestions to the authors for improvement.

Specific comments:
• Minor typographical error: Line 68: Major League Baseball should be spelled out in its entirety prior to using the MLB abbreviation.

• Referencing error: Line 434: The authors cite Passan (2016) as support for a claim that $1.5 Billion is spent annually in Major League Baseball on pitching talent. This reference is not included in the reference list. Additionally, I believe the authors are refereeing to “The Arm: Inside Billion-Dollar Mystery of the Most Valuable Commodity in Sports” By Jeff Passan (2016). This $1.5 billionfigure from the reference actually appears in the first sentence of the inside cover of the book, I suggest that the authors track that figure from the book back to its original source for the reference.

• Typographical error Lines 288-293: The authors describe a “randomized control trial” by Freeston et al. (2016) where 13 junior baseball players were tested on two occasions, seven days apart. The referenced study actually describes a “randomized crossover trail” as indicated in the title of the referenced article “Indicators of throwing arm fatigue in elite adolescent male baseball players: A Randomized crossover trial”.

• Article structure: Results section: Thank you for the authors for including the PRISMA checklist in their supplementary file. However, although the authors have included synthesis tables for kinematic (Table 2) and performance changes (Table 3) with fatigue in pitchers a more detailed synthesis table is need for a systematic review (item 20 in PRISMA checklist). I suggest combining all reviewed articles into a large table with different sections for Kinematic changes, performance changes and injuries associated with fatigue in baseball pitchers. This table should also include a risk of bias score for each reviewed article (see comment in experimental design).

Experimental design

The manuscript in its current form is more of a narrative review that used a systematic search protocol. For this to be a true systematic review the following changes should occur.

• According to the PRISMA statement by Moher et al 2009, “A systematic review is a review of a clearly formulated question that uses systematic and explicit methods to identify, select and critically appraise relevant research, and to collect and analyse data from the studies that are included in that review” (pg. 264). Although the authors have indicated on their PRISMA checklist that they have adhered to item 12 (risk of bias in individual studies). A more robust appraisal should be employed incorporating a tool to assess the quality of the studies. The Downs and Black checklist or a modified version of, is commonly used for this (Downs SH, Black N. The feasibility of creating a checklist for the assessment of the methodological quality both of randomised and non-randomised studies of health care interventions.

• A synthesis table that includes all reviewed literature and the risk of bias score (see previous comment in basic reporting sections)

Validity of the findings

Conclusions are well stated and link back to the research questions

---

## Round 0.2 · Minor Revisions

Thank you for taking the time to respond to the reviewers comments. The paper is now much better than previous. There remains a few minor edits that one reviewer would like to see. Can you please attend to these, after which a final decision will be made.

Reviewer 1 ·

Basic reporting

The changes made by the author have improved the focus and clarity of this paper.

Experimental design

The changes made by the author have improved the focus and clarity of this paper.

Validity of the findings

I think some further minor changes are needed before publication to ensure the interpretation matches the strength of the reviewed literature.

Additional comments

Thank you for responding to my previous feedback. The changes that you have made have greatly improved the manuscript. I have the following additional comments about the results and discussion that I think should be addressed before publication.

Line #334-335: that is only a 2.5 degree change in knee flexion angle which is a very small change. Please acknowledge effect magnitude not just direction i.e. “…small increase in knee flexion…” whenever you are discussing findings in the discussion section.

Table 3: please add the mean difference or pre- post- values for these significant changes so that the magnitude can be compared with the other studies. You have done this for all studies except this one.

Line #338-339 “A retrospective analysis study identified, during a game, the greatest muscle fatigue occurs in the extensor carpi radialis longus” this needs to be linked to kinematics which is the focus of the paragraph based on the subheading.

Line #345: the reference to Chalmers et al (2017) seems to be a narrative commentary on pitching mechanics and injury risk. Is there actually data to say that “slight deviations” cause injuries or is that just the author’s opinion? If the latter, please revise so that it is clear that “slight deviations” are thought to be important but such data does not exist. I’m assuming it does not exist otherwise it would have been formally included in your review.

Line #351-354: this section talks about forces but reports percentage data. What do the percentages indicate? You report one for the first and last innings so they don’t appear to indicate percentage change. Please make this clearer.

Line #396 “Fatigue linked to injury”: Only two studies (Lyman et al. 2001 and 2002) in Table 5 have findings that relate fatigue (self-reported fatigue or pitch count) to pain (none have done this with injury). Clearly there is a need for more work like this and for injuries not just pain to be considered. You need to acknowledge the lack of research into the relationship and emphasize the importance of this in the future.

Line #419: why is the Fleisig et al. (2009) study not included in your review if it reports injury incidence in injured pitchers?

Discussion: please add a limitations paragraph that highlights the lack of true fatigue and injury studies (i.e. only the Lyman et al. 2001 and 2002 studies have done this and they used pain not injury) and the fact that this review is based largely on data from junior or collegiate athletes rather than professional athletes. It is important to highlight these issues with past literature so that future studies can address them. Also explain why meta-analysis was not possible and what could be done to allow meta-analysis to be undertaken in the future. True Level 1 evidence should be the goal for this topic eventually.

·

Basic reporting

Basic reporting for the revised manuscript has been substantially improved from the original submission.

Experimental design

The methodology in the revised manuscript has been improved. The manucript demonstrates methodological rigour associated with the eligibility criteria and the risk of bias assessment.

Validity of the findings

The findings presented in the revised mansucript presented clearly. Conclusions are supported by the findings.

---

## Round 0.3 · accepted · Accept

Thank you very much for taking the time to respond to the reviewer comments. I think you will agree this is a process which ultimately improves paper quality.